

# Ensemble prediction using a new dataset of ECMWF initial states - OpenEnsemble 1.0

Pirkka Ollinaho[1], Glenn D. Carver[2], Simon T. K. Lang[2], Lauri Tuppi[3], Madeleine Ekblom[3], and Heikki Järvinen[3]

[1]Finnish Meteorological Institute (FMI), Helsinki, Finland
[2]European Centre for Medium-Range Weather Forecasts (ECMWF), Research Department, Reading, UK
[3]Institute for Atmospheric and Earth System Research / Physics, Faculty of Science, University of Helsinki, Finland

**Correspondence:** Pirkka Ollinaho (pirkka.ollinaho@fmi.fi)

**Abstract.** Ensemble prediction is an indispensable tool of modern numerical weather prediction (NWP). Due to its complex data flow, global medium-range ensemble prediction has so far remained exclusively as a duty of operational weather agencies. It has been very hard for academia therefore to be able to contribute to this important branch of NWP research using realistic weather models. In order to open up the ensemble prediction research for a wider research community, we have recreated all 50+1 operational IFS ensemble initial states for OpenIFS CY43R3. The dataset (OpenEnsemble 1.0) is available for use under a Creative Commons license and is downloadable from an https-server. The dataset covers one year (December 2016 to November 2017) twice daily. Downloads in three model resolutions ($T_L159$, $T_L399$ and $T_L639$) are available to cover different research needs. An open-source workflow manager, called OpenEPS, is presented here and used to launch ensemble forecast experiments from the perturbed initial conditions. The deterministic and probabilistic forecast skill of OpenIFS (cycle 40R1) using this new set of initial states is comprehensively evaluated. In addition, we present a case study of typhoon Damrey from year 2017 to illustrate the new potential of being able to run ensemble forecasts outside major global weather forecasting centres.

## 1 Introduction

The conventional method of predicting the future state of the atmosphere is to make a single model forecast from the analysis which is the current best state estimate of the atmosphere. Due to limitations in observations and in the data assimilation system, a measure of uncertainty remains in this state estimate. Moreover, the forecast model has its own uncertainties (see e.g. Leutbecher and Palmer, 2008). Ensemble forecast systems are designed to complement the deterministic forecast by providing a set of alternative but equally plausible future evolutions of the atmospheric state. The spread of these forecasts can be interpreted, for instance, in terms of predictability of the current state, or as alternative forecasts.



For approximately the past 25 years, ensemble forecasting research has been mostly a duty of major operational forecasting centres. Without access to an Ensemble Prediction System (EPS) itself, the majority of academic research on the topic has been limited to studying the end-products, i.e. various aspects of ready-made ensemble forecasts, or using idealized modelling setups. Although a great amount of progress has been made on learning about and improving ensemble forecasts over the years, many open research questions and technical development possibilities still exist, for example in the following areas: (1) operational forecasting centres constantly reassess current operational implementations of initial state uncertainty (see e.g. Lang et al., 2015, 2019) and model uncertainty representations (see e.g. Lock et al., 2019), and explore completely new methods (see e.g. Ollinaho et al., 2017; Leutbecher et al., 2017), (2) ensemble modeling studies of tropical storms (see e.g. Lang et al., 2012) offer a unique way to study forecast impacts of both initial state uncertainty and model uncertainty representations, (3) EPSs also provide a potent way of automating the model tuning process (Ollinaho et al., 2013, 2014; Tuppi et al., under review).

This paper describes a new dataset of ensemble initial states covering a one year period twice a day, starting from the 1st of December 2016, courtesy to European Centre for Medium-Range Weather Forecasts (ECMWF). The dataset, OpenEnsemble 1.0, is freely usable under a Creative Commons licence and downloadable from an https-server (until foreseeable future, details in Section 4). Moreover, we present a workflow manager, called OpenEPS, to run OpenIFS ensembles from the new dataset. OpenEPS is also freely available under an Apache 2.0 license.

We hope the OpenEnsemble dataset and the workflow manager will enable the wider academic research community to contribute to ensemble forecasting research with realistic modelling tools. Additional new potential will be opened also in a variety of applications. Renewable energy production is dependent on weather conditions, such as cloudiness, wind speed, and icing. Increasing demand has clearly exposed this sector to potential benefits of accurate weather forecasts. Utilizing ensemble forecasts in this context can reveal new sources of added value for the users (see e.g. Sperati et al., 2016; Rasku et al., 2020). Ensemble forecasts are also important for flood forecasting applications (Smith et al., 2016). Lastly, the available uncertainty information from ensemble forecasts has particular value for prediction of extreme weather events, such as extra-tropical and tropical storms (see Friederichs et al., 2018, and references therein).

The dataset is aimed for use with the ECMWF OpenIFS model, described in Section 2. Although the initial states are native to the OpenIFS model, there is no technical reason why they could not be used in initializing any other forecasting model as well. However, implementing such a use case would likely not be a trivial task and there would likely be a longer spin-up period affecting the forecast quality early on. We discuss the initial state perturbation strategies used by ECMWF in Section 3. The dataset and instructions on how to use it is provided in Section 4. The OpenEPS software as well as a setup for testing the dataset of initial states is described in Section 5. Performance of an OpenIFS ensemble started from the initial state perturbations is shown in Section 6.





## 2 ECMWF OpenIFS

The ECMWF Integrated Forecast System (IFS) was first used for operational forecasting in March 1994. Since that time, the IFS has been continually improved and its forecast performance assessed. The relative contribution of model improvements,
reduction in initial state error and increased use of observations to the IFS forecast performance is continuously monitored by ECMWF. Haiden et al. (2017) provides a detailed report on the IFS forecast performance for the IFS cycle 43R3, upon which the dataset described here is based. A detailed and up-to-date record of changes between IFS release cycles can be found at ECMWF (2019a).

The ECMWF OpenIFS activity, launched in 2011, provides a portable version of the IFS to ECMWF member state hydro-
meteorological services, universities and research institutes for research and education purposes for use on computer systems external to ECMWF. It is used in a wide range of studies from teaching masters level courses, to forecasting extreme events and inclusion in coupled climate models. As OpenIFS shares the same code base as IFS, the scientific forecast capability of the two models is identical and the model description in this section applies equally to IFS except where indicated. The OpenIFS model supports all resolutions up to the ECMWF operational resolution and ensemble forecast capability. The ocean model,
data assimilation and observation handling components of IFS are not included in OpenIFS. A detailed scientific and technical description of IFS, applicable to OpenIFS, can be found in open access scientific manuals available from the ECMWF website (ECMWF, 2019b).

The OpenIFS model is a global model and uses a hydrostatic, spectral, semi-Lagrangian dynamical core for all forecast resolutions, with prognostic equations for the horizontal wind components (vorticity and divergence), temperature, water vapour
and surface pressure. The horizontal resolution is represented by both the spectral truncation wavenumber (the number of retained waves in spectral space) and the resolution of the associated Gaussian grid. Model resolutions are usually described using a Txxx notation where xxx is the number of retained waves in spectral representation. An additional letter is used to describe the type of physical grid associated with the spectral truncation: $T_L$ is used to denote so-called 'linear' grids where the maximum wavenumber (shortest wave) is represented by the spacing between two adjoining gridpoints; $T_{CO}$ is used to denote
the cubic-octahedral grid where the maximum wavenumber is represented by four gridpoints. Both these grids use a reducing number of gridpoints along lines of latitude approaching the poles (see Malardel et al., 2016, for more details). OpenIFS based on IFS cycle 43R3 is the first with the capability of using the $T_{CO}$ horizontal grid. The vertical resolution varies smoothly with geometric height, and is the finest in the planetary boundary layer becoming more coarse towards the model top.

For a description of OpenIFS physical parametrizations we refer the reader to the ECMWF online documentation (ECMWF,
2019b). Although OpenIFS contains the ECMWF wave model (Janssen, 2004), the wave model is not used in the experiments here. The OpenIFS model also includes stochastic parametrization schemes of IFS to represent model error (see Leutbecher et al., 2017, for an overview).





## 3 Initial state perturbations in the ECMWF ensemble

### 3.1 Singular Vectors

Singular Vectors represent the fastest-growing perturbations to a weather forecast - called the trajectory - within a finite time interval (Lorenz, 1965; Buizza, 1994; Palmer et al., 1998). In order to compute singular vectors one linearises the governing equations around a given trajectory. The idea behind using singular vector based initial perturbations is that these are the dynamically most relevant structures and hence will dominate the forecast uncertainty (Ehrendorfer and Tribbia, 1997; Leutbecher and Palmer, 2008; Leutbecher and Lang, 2014). Here, growth is defined with respect to a specific metric. The metric

used at ECMWF is the so called dry total energy norm (Leutbecher and Palmer, 2008) and singular vectors are computed with an optimisation interval of 48 h and a spatial resolution of $T_L42$ and 91 vertical levels. Different sets of singular vectors are computed, the leading 50 singular vectors for the Northern and Southern Hemisphere, and the leading 5 singular vectors for each active tropical cyclone (see Buizza, 1994; Puri et al., 2001; Barkmeijer et al., 2001; Leutbecher and Palmer, 2008, for details). While singular vectors targeted at the extra-tropics are optimised for the whole troposphere, singular vectors targeted

at tropical cyclones are optimised for growth below 500 hPa.

### 3.2 Ensemble of Data assimilations

At ECMWF an ensemble of 4D-Var data assimilations (EDA, Buizza et al., 2008b) is run to provide uncertainty estimates for both the ensemble forecasts and the high resolution analysis. The EDA consists of a number of perturbed members and one unperturbed control member. Originally it was run with 10 perturbed members but this changed in 2013 when the number of

members was increased to 25. Recently the number of members was increased again to 50 (Lang et al., 2019). In this study we make use of the EDA configuration with 25 perturbed members. For each EDA member the observations and boundary conditions are perturbed and the SPPT scheme is used to simulate the impact of model error.

### 3.3 Construction of initial perturbations

Perturbations based on the EDA short-range background forecasts are combined with singular vector based perturbations to

build the initial conditions for the ECMWF ensemble forecasts. EDA perturbations are derived by subtracting the mean of the EDA from each EDA member, resulting in a total of 25 perturbations. In addition to the EDA perturbations, 25 singular vector based perturbations are generated.

For each singular vector perturbation an individual linear combination from all singular vectors is constructed. This is done by scaling the singular vectors by random numbers drawn from a multi-variate Gaussian distribution (see Leutbecher and

Palmer, 2008; Leutbecher and Lang, 2014).

The perturbations are centred on the high resolution analysis and have a plus-minus symmetry, i.e. the initial conditions of the first perturbed ensemble forecast member are derived by adding the first EDA perturbation and the first singular vector perturbation to the high resolution analysis. The initial conditions of the second perturbed ensemble forecast member are then



obtained by subtracting the first EDA perturbation and the first singular vector perturbation from the high resolution analysis, and so on. Thus the initial perturbations of the even ensemble members have the opposite sign to the perturbations of the odd ensemble members. This setup makes it possible to distribute the 25 EDA perturbations between the 50 ensemble forecast members. The plus-minus symmetry ensures that the mean of the perturbed analyses equals the unperturbed control analysis. Note that this is no longer the case from 2019 onwards in the operational ECMWF system. Following the introduction of the 50 member EDA the plus-minus symmetry is not required anymore (Lang et al., 2019).

## 4  The dataset

Our dataset of ensemble initial states covers a one-year period from December 1st 2016 to November 30th 2017. The initial states have been generated closely following what is done in the ECMWF operational ensemble. We use IFS cycle 43R3, which was operational from July 11th 2017 to June 5th 2018, to generate the ensemble initial states. The single major difference to the operational ensemble setup is that the highest model resolution is $T_L639$ (instead of $T_{CO}639$). This model version is also the basis for the OpenIFS release CY43R3v1.

The most relevant changes between this model version and the currently operational ECMWF version (CY46R1) that affect the ensemble initial states arise from changes to the model uncertainty representations (see Lock et al., 2019) and from the removal of the plus-minus symmetry. On top of this list, a number of changes have been introduced to the IFS model, the IFS data-assimilation process and amount of observations used in data-assimilation. These all will naturally also affect the ensemble initial states, due to direct or indirect contributions.

The data is tarred, gzipped and packed such that a single tarz-file contains all initial states (control state and perturbed initial states) for a given date and time (00 and 12UTC). The files are in the form "YYYYMMDDHH.tarz". The data is furthermore arranged into separate directories for the three available resolutions:

1. https://a3s.fi/oifs-t159/YYYYMMDDHH.tgz (∼1.4GB per file)

2. https://a3s.fi/oifs-t399/YYYYMMDDHH.tgz (∼7.9GB per file)

3. https://a3s.fi/oifs-t639/YYYYMMDDHH.tgz (∼19.5GB per file)

Table 1 lists the types of files a single downloaded tarz-file consists of. In each file there are 50 of the pan, psu, pua and pert files, one for each perturbed initial state. The first four files in Table 1 form an unperturbed atmospheric initial state that is used to initialize a control forecast. ICMGGoifsINIUA-files contain model level variables on a Gaussian reduced grid. ICMSHoifsINIT-files contain model level variables in spherical harmonics representation. ICMGGoifsINIT-files contain a number of surface level variables on a Gaussian reduced grid. ICMCLoifsINIT-files contain climatological surface fields for albedo at different radiation wave lengths, leaf area indexes, soil temperature and sea ice area fraction. For detailed description of the contents of each file we refer the reader to Appendix A. On top of these four files, OpenIFS requires static climatological files for radiation calculations and various namelists describing the hybrid sigma coordinates etc. In order to utilize the wave





model, a separate set of wave model input files and namelists is required. These can be obtained from OpenIFS-support team on request.

**Table 1.** Files contained in a tarz-file for a given date and time. ${RES} is replaced with the spherical truncation number, i.e. 159/399/639.

| File name | Use as | Description |
|---|---|---|
| ggml${RES} | ICMGGoifsINIUA | Model level variables on a Gaussian grid |
| ICMCLoifsINIT.1 | | Climatology file |
| ICMGGoifsINIT | | Surface level variables on a Gaussian grid |
| ICMSHoifsINIT | | Model level variables in Spherical harmonics |
| pan_001-pan_050 | ICMGGoifsINIUA | |
| psu_001-psu_050 | ICMGGoifsINIT | |
| pua_001-pua_050 | ICMSHoifsINIT | |
| pert_001-pert_050 | | Raw SV perturbation fields |

   The last four files in Table 1 contain different forms of initial state perturbations. pan_${MEMBER} can be used to replace ICMGGoifsINIUA and are constructed as high resolution analysis +/- EDA perturbations. pua_${MEMBER} can be used to replace ICMSHoifsINIT and contain the final spherical harmonics representation of an initial state containing both EDA and SV

perturbations. psu_${MEMBER} can be used to replace ICMGGoifsINIT files and contain initial states with EDA pertubations. And finally, the pert_${MEMBER} files contain raw Singular Vector perturbations. These can be used to decompose the initial state perturbations into SV and EDA parts.

   These files can be used to form four different kinds of initial states for experimentations, listed in Table 2. Control forecast and SV+EDA perturbed forecasts can be initialized directly from the provided files. For EDA only or SV only perturbations

some file manipulation is required. In order to get initial states with EDA only perturbations, pert-file needs to be subtracted from pua-file. An example on how to do this using Climate Data Operators (CDO; Schulzweida, 2019) software is given in Appendix B. The file manipulation is always subtraction: the +/- symmetry is build into the files, i.e. pert_001 and pert_002 will be identical fields with different signs. For SV only perturbations, the control state should be used, and the pert-files added to the ICMSHoifsINIT-files. Again, the same procedure with grid point conversion and +/- symmetry applies here. The workflow

shown in Appendix B can be applied here with minor modifications.

## 5   Running ensembles from the dataset

### 5.1   Workflow manager - OpenEPS

In order to run large ensembles of a forecast model, a workflow manager is essential. For this purpose, we use here a simple yet efficient software called OpenEPS. We want to emphasize that the provided dataset of initial states is not tied to this, or any,

software. OpenEPS is mostly written in bash but utilizes GNU make to handle parallel job executions in HPC, Linux cluster or laptop environments. The software is freely available under an Apache 2.0 license (https://github.com/pirkkao/OpenEPS).





**Table 2.** Types of initial states that can be constructed from the dataset.

| Type | Files needed | Manipulation |
|---|---|---|
| All | ICMCLoifsINIT.1 | |
| Control/deterministic forecast | ICMGGoifsINIUA | ggml${RES} |
| | ICMGGoifsINIT | ICMGGoifsINIT |
| | ICMSHoifsINIT | ICMSHoifsINIT |
| EDA+SV perturbations | ICMGGoifsINIUA | pan_001 |
| | ICMGGoifsINIT | psu_001 |
| | ICMSHoifsINIT | pua_001 |
| EDA only | ICMGGoifsINIUA | pan_001 |
| | ICMGGoifsINIT | psu_001 |
| | ICMSHoifsINIT | pua_001 - pert_001 |
| SV only | ICMGGoifsINIUA | ggml${RES} |
| | ICMGGoifsINIT | ICMGGoifsINIT |
| | ICMSHoifsINIT | ICMSHoifsINIT + pert_001 |

Instructions on how to use the software, as well as a few example cases, are provided with the software download. We will nonetheless provide a concise description of the OpenEPS software here as the workflow would be similar with any workflow manager. The general workflow is handled as follows:

0. Setup a computing environment specific file containing various architecture settings

   1. Choose experiment specifications (resolution, number of ensemble members, computing resources, etc.)

   2. Run OpenEPS. This will:

      – Construct required path-structure for the experiment

      – Generate run configurations (fort.4) for OpenIFS

– Link full initial states or generate a set of initial states from the dataset (SV only, EDA only, or change amplitudes of the perturbations) for the first ensemble initialization date

      – If run on an HPC, reserve run resources for the model execution and submit the batch job

   3. Run model forecasts for the given date with the available resources

   4. Once all ensemble member forecasts are complete, execute one or more of the following

– Post-process model outputs

      – Run additional tasks for e.g. algorithmic model tuning





5. Link or generate initial states for the next date

6. Go back to 3 until all dates have been cycled through

Step 0 only needs to be completed once for each unique computing environment. In the current HPC implementation,
OpenEPS reserves all the wanted computing resources at the same time. For example, if the user wants to run a total of 5
ensemble members, each to be executed concurrently and each using 20 cores would mean submitting a batch job requesting
a reservation of 100 cores for a timeslot of N minutes. If the user would want to run 50 ensemble members in total following
the previous setup, OpenEPS would compute these in 10 consecutive batches within the same batch reservation. Instead of
reserving 100 cores for N minutes, the cores would now be reserved for N*10 minutes. One could naturally increase the
amount of computing resources as well in order to keep the execution time to a minimum, i.e. reserve 100*10 cores for N
minutes.

It is also possible to do online post-processing within the workflow, i.e. run scripts to manipulate each model forecast after
they are finished. Note that due to the nature of the HPC implementation this means all the reserved resources might be sitting
idle while this is happening. Usually the computing resources required for model forecast calculations are much larger than
those used in post-processing the output, so caution is advised here. We recommend using this option only when online post-
processing is required as part of the workflow, e.g. in algorithmic model tuning. Also, since manipulation of the initial state
files is resource demanding, it it is highly advisable that modifications to the initial states (separation of SV and EDA parts for
example) are done as a separate task before the actual model integrations. Workflow for this is also supported in OpenEPS.

### 5.2 Forecast model setup

Although the initial states have been generated using IFS version CY43R3, we use here the OpenIFS version matching IFS
CY40R1 as the forecast model. This is due to practical reasons: at the time of writing this paper, the matching cycle for OpenIFS
(CY43R3v1) was still in preparation. We foresee that the forecast model difference will affect the testing somewhat, mainly due
to differing analysis and forecast biases, i.e. the analysis bias is affected by the model cycle and hence a CY43R3 analysis and
CY43R3 forecast model will have more similar biases than a CY43R3 analysis and a CY40R1 forecast model. This will result
in potentially better scores when analysis and forecast were created by the same cycle. The forecasting skill in early forecast
lead times might also be somewhat degraded due to a stronger-than-usual spin-up effect. However, we still feel confident
that the forecast skill evaluation is of value despite the forecast model differences. A number of physical parametrization and
model dynamics changes happened between CY40R1 and CY43R3, we refer interested readers to the ECMWF OpenIFS and
IFS webpages (ECMWF, 2020a, 2019a, b).

### 5.3 Experiment setup


We have run a number of experiments in order to assess how well the initial state perturbations fare with OpenIFS, these
are listed in Table 3. The experiments cover the three provided horizontal resolutions: $T_L 159$ ($\sim 120\,\mathrm{km}$), $T_L 399$ ($\sim 50\,\mathrm{km}$)
and $T_L 639$ ($\sim 32\,\mathrm{km}$). Also, the different initial perturbation types (SV and EDA) were tested separately in order to illustrate





the efficiencies of the perturbations in generating ensemble spread with the OpenIFS setup. All of the experiments were run
without any model uncertainty representations.

**Table 3.** Experiments conducted for this study. Experiment name includes the used resolution as well as the type of initial state perturbations used. Ensemble size, the number of ensemble initialization dates and finally notes regarding the experiment.

| Name | Ens size | Dates | Notes |
| --- | --- | --- | --- |
| $T_L159$-SV | 50 | 46 | SV pert only |
| $T_L159$-EDA | 50 | 46 | EDA pert only |
| $T_L159$-BOTH | 50 | 46 | EDA and SV pert |
| $T_L159$-SV+ | 8 | 46 | SV pert multiplied by 1.2 |
| $T_L159$-EDA+ | 8 | 46 | EDA pert multiplied by 1.2 |
| $T_L159$-BOTH+ | 8 | 46 | EDA and SV pert multiplied individually by 1.2 |
| $T_L399$-SV | 20 | 46 | SV pert only |
| $T_L399$-EDA | 20 | 46 | EDA pert only |
| $T_L399$-BOTH | 50 | 46 | EDA and SV pert |
| $T_L639$-SV | 20 | 46 | SV pert only |
| $T_L639$-EDA | 20 | 46 | EDA pert only |
| $T_L639$-BOTH | 20 | 46 | EDA and SV pert |

Running large numbers of ensemble forecasts requires a substantial amount of computational resources. Leutbecher (2017) demonstrated that the number of ensemble start dates is much more important than the size of the ensemble in extracting the mean probabilistic skill of the system. Thus, we keep the number of start dates high, but decrease the ensemble size for the higher resolutions in order to save computational resources: only the $T_L159$ experiments and the basic setup for $T_L399$ have
been run with the full 50 ensemble members. $T_L399$ experiments testing the effect of the initial state perturbation methods individually have been run with a reduced ensemble size of 20 members, as have all the $T_L639$ experiments. We will showcase later in Section 6.2 that by using fair scores the ensemble size chosen here is indeed more than enough to extract the probabilistic skill of the system.

An additional set of experiments with $T_L159$ resolution has also been run where the amplitudes of SV and EDA perturbations
have been inflated by multiplying the perturbation fields with a constant number. This exercise aims to demonstrate how the initial state amplitudes can be used to tune the ensemble skill. In the combined perturbations experiment (BOTH+), both of the perturbation types have been increased individually and then added together. These experiments use an ensemble size of 8 members.

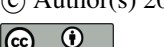



# 6 Ensemble forecast skill evaluation

## 6.1 Ensemble mean RMSE and ensemble spread

It is common to assess the skill of an ensemble by calculating the ensemble mean root mean squared error (RMSE) and the
ensemble standard deviation (ensemble spread). The former measures how accurate the ensemble mean is, i.e. how near the
mean of the ensemble forecasts is from analysis fields or observations. The latter verifies whether the ensemble forecasts
simulated wide enough range of possible atmospheric states to reflect the error characteristics of the ensemble mean. Ideally,
one would want the ensemble mean RMSE to be as small as possible and the spread to be equal to the ensemble mean RMSE
on average over many cases and within sampling uncertainty caused by a finite number of cases and ensemble members (see
Leutbecher and Palmer (2008), for an in-depth discussion). We use here operational ECMWF analyses from the forecast period
as the truth. The model output is truncated to a 1°/1° regular grid before any other post-processing is done.

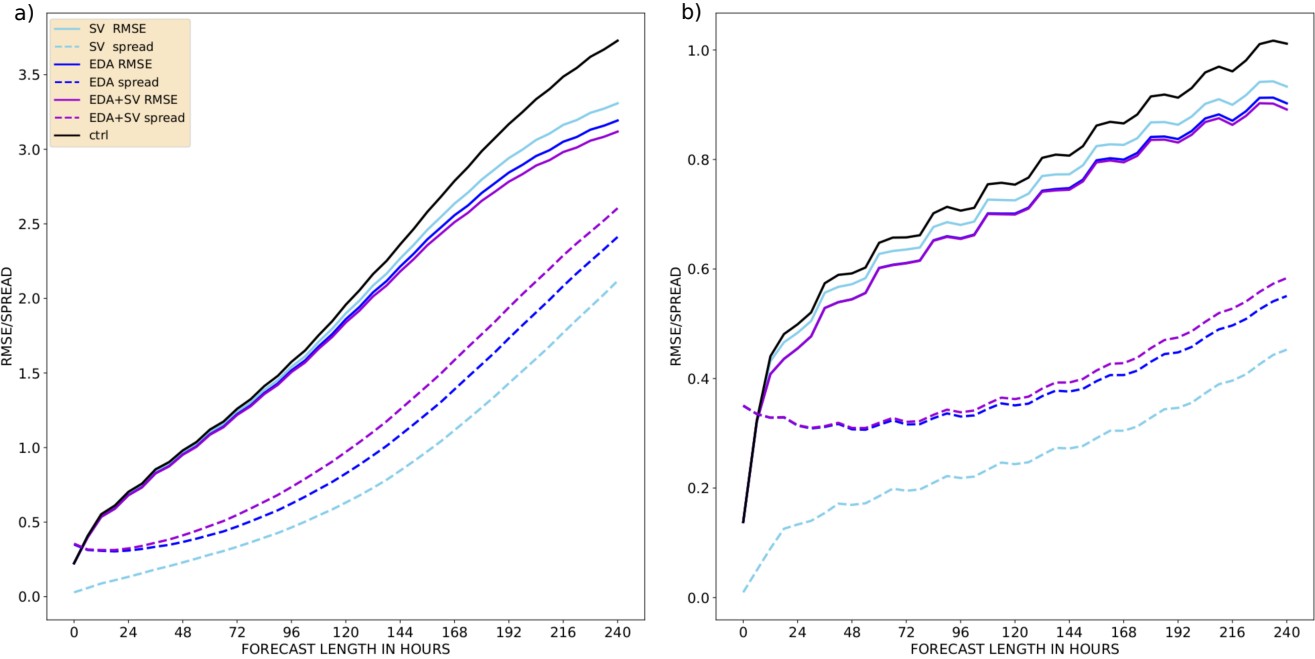

**Figure 1.** Ensemble mean RMSE (solid) and ensemble spread (dashed) of temperature at 850 hPa as a function of forecast lead time up to
h. $T_L159$ model resolution with 50 ensemble members. Mean over 46 start dates. Experiments included: only SV perturbations (cyan),
only EDA perturbations (blue), and both SV and EDA perturbations (violet). Also shown here is the unperturbed control member RMSE
(black). Northern hemisphere (a) and tropics (b).

In Figure 1 the ensemble mean RMSE (solid) and ensemble spread (dashed) are shown for the first three $T_L159$ experiments
(SV, EDA, BOTH). The left-hand side panel represents scores for the Northern Hemisphere (NH) and the right-hand side for
the Tropics (TR). The EDA perturbations produce more ensemble spread than the SV perturbations in both NH and TR. It



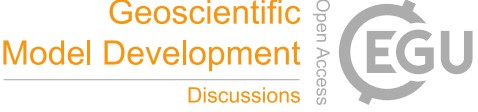

is also quite evident that including both of the perturbations further improves the ensemble spread, i.e. it moves closer to the ensemble mean RMSE. Both perturbations also improve the forecast skill of the system when compared against a control run without any perturbations (black line). Interestingly, the EDA perturbations do not really start to grow before forecast length

of 48 h in NH and 96 h in TR. The added benefit of having both of the perturbation types active at the same time can also be observed in the mean forecast skill beyond forecast leadtimes of 5 (8) days in NH (TR). The behaviour w.r.t. the types of initial state perturbations is similar at resolutions $T_L 399$ and $T_L 639$ (not shown).

An increased horizontal resolution leads to a much improved forecast skill of the system, as can be seen in Fig 2 where the experiments with both SV and EDA perturbations active are plotted for all three resolutions. For both $T_L 399$ and $T_L 639$

resolutions this is due to both a larger spread and an improved forecast skill of the ensemble. Note that there is a sampling difference between the two lower resolution experiments and the $T_L 639$ experiment: the former two cases have 50 ensemble members whereas the latter one has 20. Ideally, one should account for the finite number of members when comparing ensemble spread and error (see Leutbecher and Palmer, 2008).

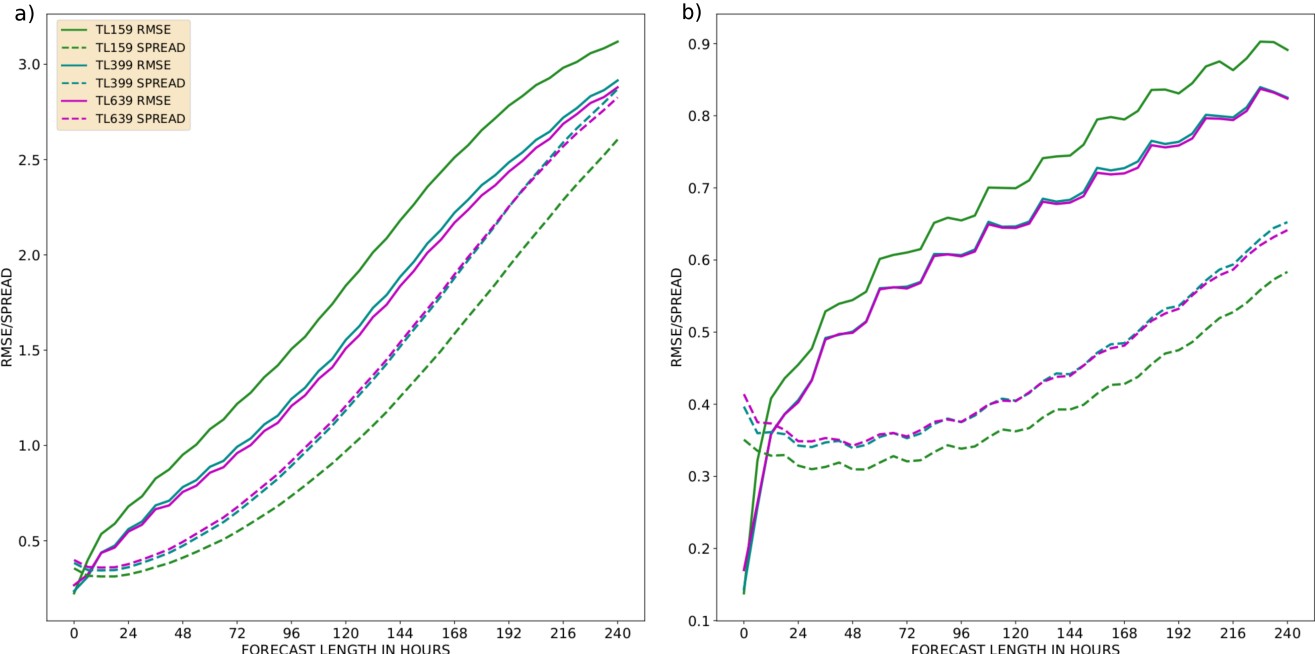

**Figure 2.** Ensemble mean RMSE (solid) and ensemble spread (dashed) of temperature at 850 hPa as a function of forecast lead time up to 240 h. Mean over 46 start dates. All experiments contain SV and EDA perturbations: $T_L 159$ (green; 50 members), $T_L 399$ (teal; 50 members), and $T_L 639$ (violet; 20 members). Northern hemisphere (a) and tropics (b).





## 6.2 Fair CRPS

Fair versions of probabilistic skill scores indicate how the system would have scored if it had had an infinite number of ensemble members[1]. Leutbecher (2017) illustrated how a fair version of the continuous ranked probability score (CRPS) can be constructed, as well as explored how many ensemble members are required in order to calculate a representative fair-CRPS. The recommended ensemble size was set to be four to eight members for scientific testing. Fig 3 shows normal CRPS (a) and fair-CRPS (b) calculated for ensembles of various sizes. The smaller ensembles are constructed from the 50-member ensemble.

A mathematical prerequisite for calculating fair-CRPS is that the ensemble members need to be exchangeable, which is not fulfilled in the ECMWF ensemble under this initial state construction style, where the SV and EDA perturbations are used with +/- symmetry (see Leutbecher, 2017). Note, this is no longer the case from 2019 onwards in ECMWF operational ensemble configuration. Therefore, the smaller sized ensembles here are either constructed from odd or even ensemble members, which fulfills the prerequisite. As per construction, the normal CRPS scores the better the more ensemble members the ensemble

contains (Fig 3). But, the fair-version of the score gives near identical results for the different ensemble sizes. This allows us to meaningfully compare different configurations.

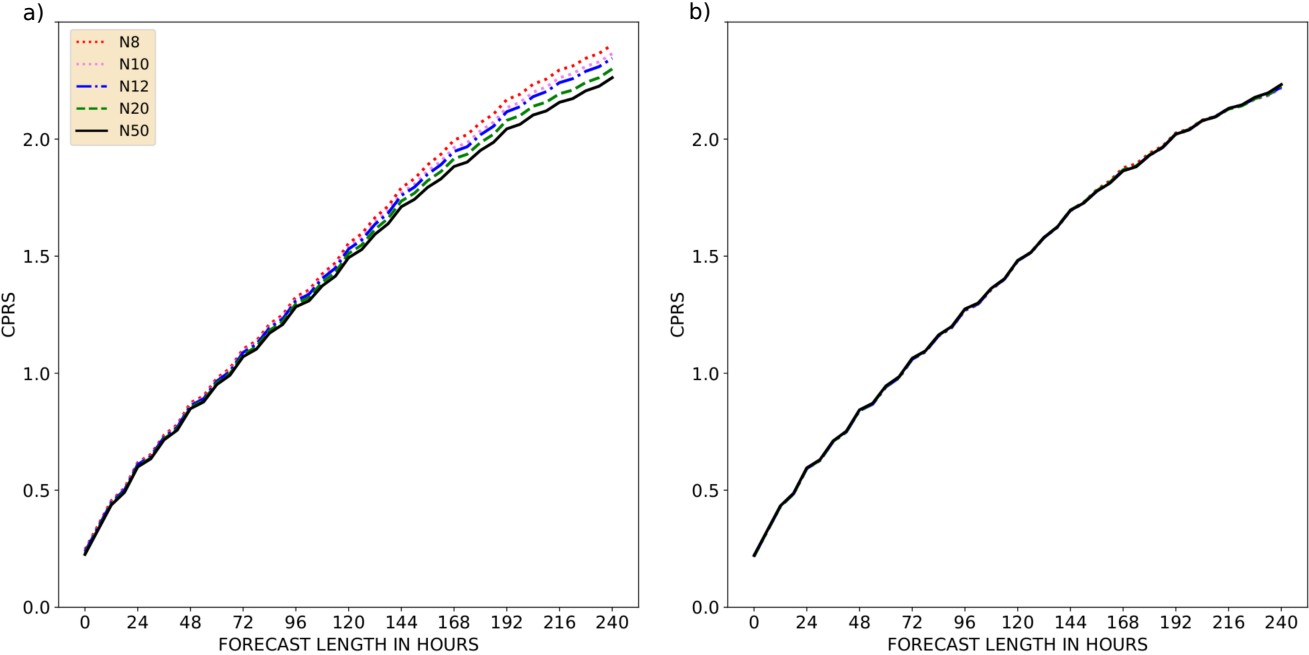

**Figure 3.** CPRS (a) and fair-CRPS (b) of temperature at 850 hPa as a function of forecast lead time, up to 240 h. Mean over 46 start dates. $T_L 159$ resolution scores for the Northern hemisphere. Different colours and linestyles represent various ensemble sizes used in calculating the scores: 50 (continous black), 20 (dashed green), 12 (dotted dashed blue), 10 (dotted pink), 8 (dotted red).

[1] In statistical terms, a fair score is a score that evaluates the underlying distribution from which the ensemble members are a random draw (Ferro, 2014)





Figure 4 shows fair-CRPS for all the experiments using standard initial state perturbation amplitudes. Using only SV perturbations generates less spread than using only EDA perturbations, this is inline with findings in Buizza et al. (2008a) and Lang et al. (2012). Noticeably, $T_L159$ resolution with EDA perturbations scores better in the Tropics than $T_L639$ with only SV per-
turbations active. The SV perturbations in the Tropics consist only of perturbations around active Tropical Cyclones, thus the relatively high fair-CRPS in the Tropics is expected. Nonetheless, having SV perturbations active on top of EDA perturbations brings clear value to all the resolutions in both NH and TR.

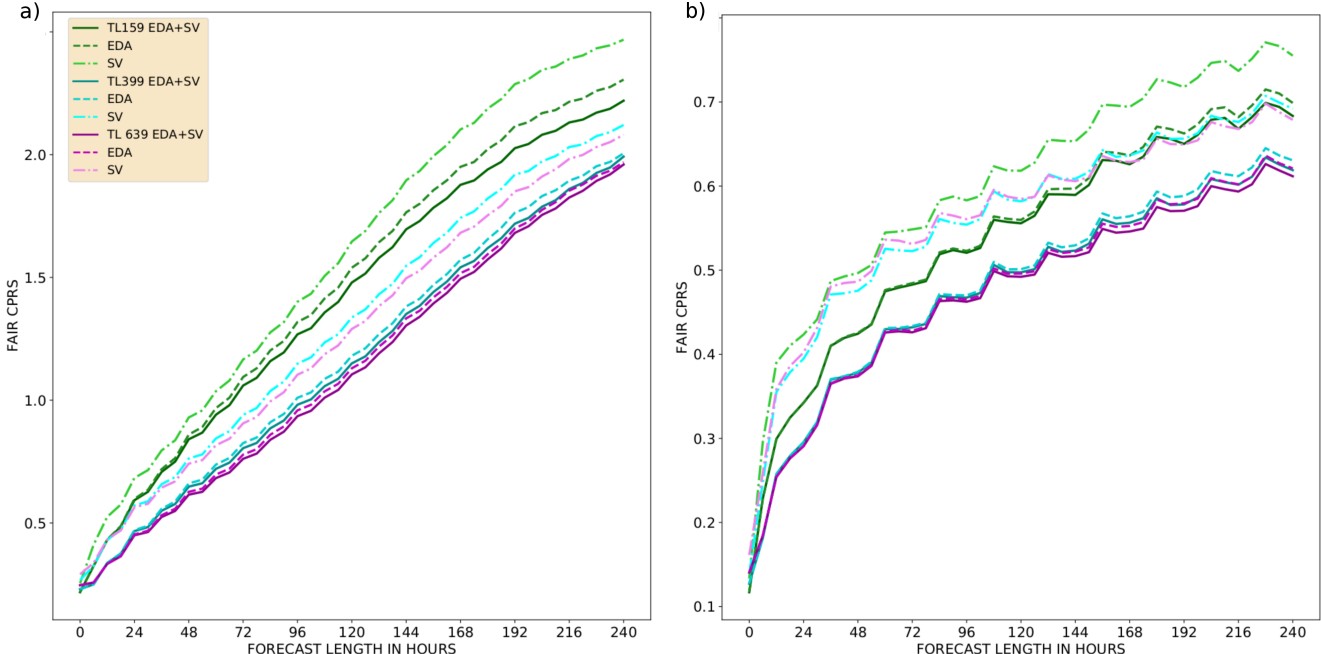

**Figure 4.** Fair-CRPS as a function of forecast lead time, up to 240 h. Mean over 46 start dates. Scores for 8-member ensembles with various resolutions and perturbation methods: $T_L159$ (green), $T_L399$ (cyan), $T_L639$ (violet), SV perturbations only (dotted dashed), EDA perturbations only (dashed), and both SV and EDA perturbations (solid). Northern hemisphere (a) and Tropics (b).

The SV perturbation amplitude is a tuning parameter of the ensemble (Leutbecher and Palmer, 2008; Leutbecher and Lang, 2014). Figure 5 illustrates the sensitivity of $T_L159$ resolution ensemble to a change in the initial perturbation amplitude. There
is a noticeable increase of skill beyond a 24 h (48 h) forecast lead time in NH when increasing the EDA (SV) perturbation amplitudes by a factor of 1.2. In TR the increase of skill becomes noticeable beyond forecast lead time of 96 h (120 h) for EDA (SV) perturbations.

We have focused on showing the forecast skill of temperature at 850 hPa over the Northern Hemisphere and the Tropics. The results from other model variables (geopotential at 500 hPa, winds at 200 hPa and 850 hPa) show very similar behaviour (not
shown). Also the Southern Hemisphere forecast skill is very much like that in the Northern Hemisphere (not shown).





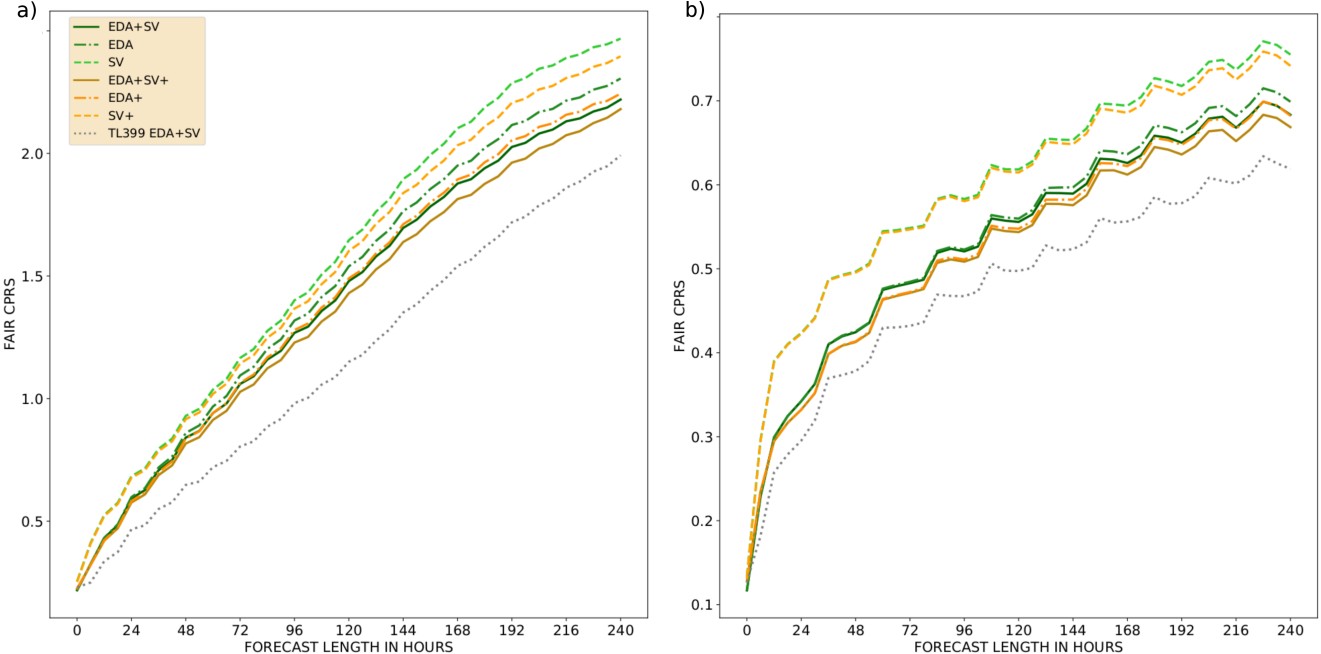

**Figure 5.** Fair-CRPS as a function of forecast lead time, up to 240 h. Mean over 46 start dates. Scores for 8-member ensembles with $T_L 159$ resolution using various perturbation methods: SV perturbations only (dashed), EDA perturbations only (dotted dashed), and both SV and EDA perturbations (solid). Normal amplitude initial state perturbations (green) and initial state perturbations increased by a factor of 1.2 (orange/brown). $T_L 399$ experiment with both SV and EDA perturbations (black dashed) drawn for reference. Northern hemisphere (a) and Tropics (b).

## 7 Applications in case studies: an example from forecasting a tropical cyclone

The following case study is aimed to illustrate the usefulness of the new data set and the OpenEPS repository. We want to emphasize that the goal here is not to dissect how the ensemble behaved or what caused the differences in the forecasts, but to give an idea how running ensembles can provide plenty of insights normally unavailable from a single model forecast.

Typhoon Damrey started as a tropical depression in the Philippine archipelago on October 31st, 2017. After moving across the open sea to the west of Philippines, it started to rapidly intensify and reached its peak strength on the third of November (the control forecast initial state for MSLP and 200 hPa wind vectors for November 2nd 12UTC is illustrated in Appendix C Fig C1). The typhoon made a landfall in Vietnam the following day and caused severe damage and loss of life (see e.g. GFDRR, 2018). Report of the operational forecasting performance of the event can be found from the ECMWF Severe Event Catalogue 290 (ECMWF, 2020b).

    We use our dataset to launch a 20-member OpenIFS ensemble starting from November 2nd 12UTC with both SV and EDA perturbations active with $T_L 639$ resolution. The ensemble mean MSLP and ensemble spread for the 0th time step of the ensemble forecast is shown in Fig 6. The observed track of Damrey is also plotted (red). Notably, the largest differences in





the initial states are focused on the South side of the typhoon core (MSLP minimum). Large differences can also be observed
in the East-West structure of the typhoon. Lang et al. (2012) show how especially the SV perturbations can rapidly alter the
TC location and intensity. Model forecast differences (due to the initial state perturbations) after a 36 h forecast lead time are
shown in Fig 7. The ensemble mean as well as majority of the ensemble members place the landfall location too South and
propagate the typhoon core too slowly (too Easterly location). The exact location and timing of the typhoon landfall is however
within the likely solutions from the ensemble.

These kind of case studies using ensembles could be used to study various mechanics of the model: Was there already
something incorrect in the TC structure in the unperturbed initial state? Did some initial state perturbations correct this and
made an impact on the TC forecast? Were local perturbations to the typhoon core essential, or was it perhaps the mean flow
forcing that was more important to get correct? Would activation of the OpenIFS wave model improve all of the forecasts due
to improved representation of momentum exchange between ocean surface and atmosphere?

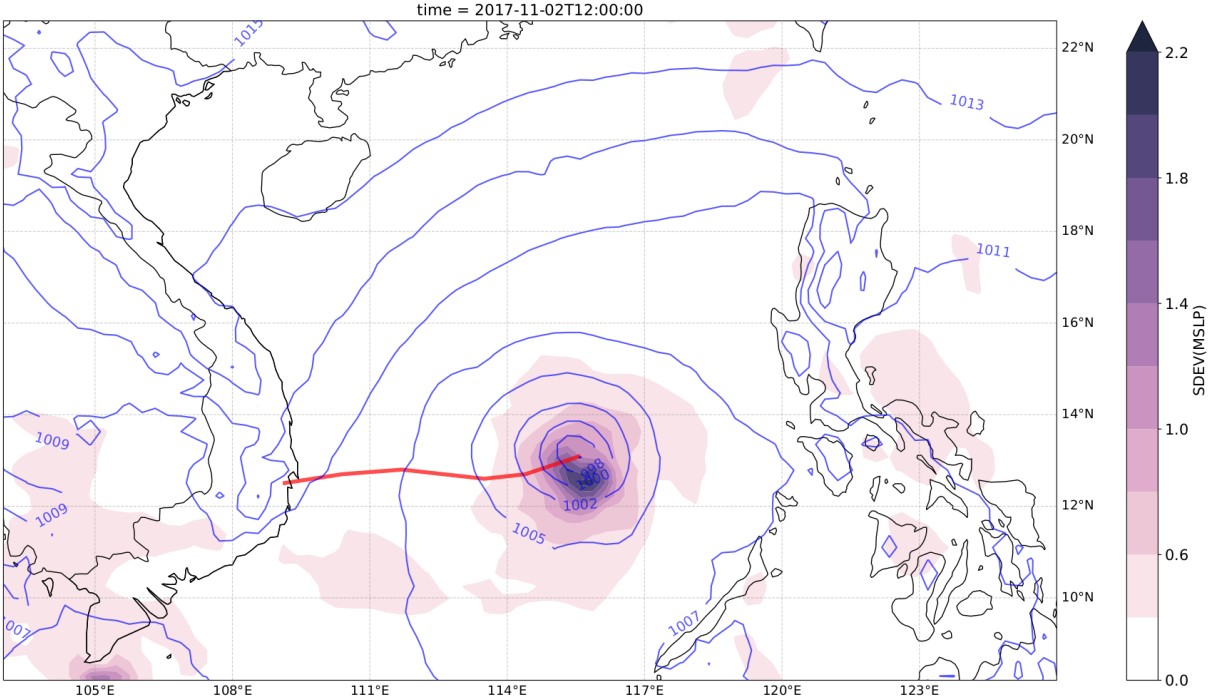

**Figure 6.** Ensemble mean MSLP (violet contour lines) and ensemble spread (coloured contours). $T_L$639 resolution. 0 h forecast initialized
on November 2nd 12UTC. Observed Damrey track between November 2nd 12UTC and 4th 00UTC (red line).

## 8 Discussion and Conclusions

We have introduced in this paper a dataset of ensemble initial states covering one year period between 1st December 2016
and 30th November 2017. The initial states have been generated closely following what is done operationally in the ECMWF



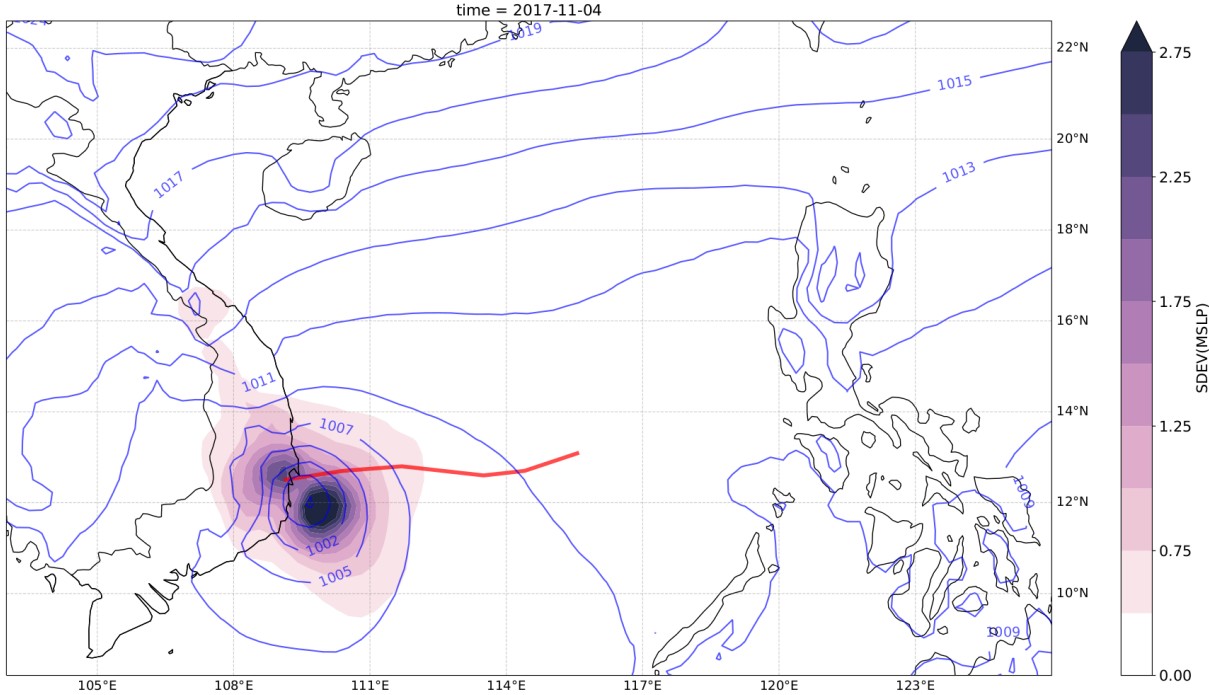

**Figure 7.** Ensemble mean MSLP (violet contour lines) and ensemble spread (coloured contours). $T_L639$ resolution. 36 h forecast initialized on November 2nd 12UTC. Observed Damrey track between November 2nd 12UTC and 4th 00UTC (red line).

ensemble and are based on ECMWF IFS cycle 43R3. Three horizontal resolutions are provided for 50+1 ensemble members: $T_L159$ ($\sim 120$ km), $T_L399$ ($\sim 50$ km) and $T_L639$ ($\sim 32$ km). The provided files can be used to construct three types of initial

310    states: (1) both SV and EDA perturbations (as in the operational ECMWF ensemble), (2) only SV perturbations, and (3) only EDA perturbations. The dataset is available for download from an https-server under a Creative Commons license.

In order to showcase the average forecast skill of the dataset, we have run forecast experiments covering all three horizontal resolutions and all three different initial perturbation types. The experiments have been run with OpenIFS CY40R1 (the newest OpenIFS version available at the time of writing). We have used here an open-source workflow manager, called OpenEPS, to

315    manage the ensemble workflow on an HPC.

For all resolutions, SV perturbations generate least spread in the ensemble. Nonetheless, having SV perturbations active on top of EDA perturbations clearly brings value to the forecast skill of the system. All perturbation types increase the accuracy of the ensemble mean when compared against a control forecast initialized from an unperturbed analysis state. We have also tested the impact of inflating the amplitudes of the initial state perturbations. Increasing the amplitudes of the initial state

320    perturbations result in an increase of the forecast skill of the system.

Inspection of especially the lowest resolution experiments reveals that all the ensembles are underdispersive, i.e. the ensemble spread is much smaller than the ensemble mean RMSE. Also, the EDA perturbations in the Tropics are inefficient in generating spread in forecast lead times less then 96 h. This is expected since our ensemble configuration is missing a model





uncertainty representation. In operational ensemble configuration having one or more model uncertainty representations has been essential in both improving the accuracy of the ensemble mean as well as increasing the spread of the ensemble. Assessing the ensemble skill when one or multiple model uncertainty representations is active on top of the initial state perturbations is something the authors will proceed to work on next. The OpenIFS release based on IFS CY43R3 includes the stochastically perturbed parameterization tendencies (SPPT) scheme (Buizza et al., 2008a) but also an early version of the stochastically perturbed parametrizations (SPP) scheme (Ollinaho et al., 2017). To assess the skill of ensemble forecasts it is also important to take biases, analysis uncertainty and observation errors into account (Yamaguchi et al., 2016). This is something we plan to do in the future.

We have also briefly demonstrated the potential of using ensemble forecasts in case studies. Typhoon Damrey, which caused severe damage in Vietnam in 2017, was simulated by generating a 20-member ensemble with $T_L639$ resolution initialized from our dataset of initial state perturbations.

We hope the the meteorological research community will find this dataset and the OpenEPS repository useful in striving towards more realistic experimentation in ensemble forecasting.

*Code and data availability.* A software licensing agreement with ECMWF is required to access the OpenIFS source distribution: despite the name it is not provided under any form of open source software license. License agreements are free, limited to non-commercial use, forbid any real-time forecasting and must be signed by research or educational organisations. Personal licenses are not provided. OpenIFS cannot be used to produce nor disseminate real-time forecast products. ECMWF has limited resources to provide support, so may temporarily cease issuing new licenses if deemed difficult to provide a satisfactory level of support. Provision of an OpenIFS software license does not include access to ECMWF computers nor data archive other than public datasets.

Other ECMWF software required for use with OpenIFS, such as ecCodes, are available as open source software using the Apache2 license and can be downloaded from the ECMWF github repository (see: https://github.com/ecmwf ).

OpenEPS software is freely available under an Apache 2.0 license. The version used in this paper can be downloaded from https://zenodo.org/badge/latestdoi/85830753. The latest development versions are available through https://github.com/pirkkao/OpenEPS.

Post-processing scripts used for this study can be found from https://zenodo.org/badge/latestdoi/290431698. The scripts for calculating the skill scores and plotting are also available https://zenodo.org/badge/latestdoi/169100276.

The dataset described here will be available until the unforeseeable future through the https-server described in this paper. The ECMWF analysis states used to calculate the skill scores are available through ECMWF MARS-archive for registered users, these can also be made available upon request from the corresponding author.





# Appendix A: Variables contained in initial state files

**Table A1.** Surface level variables on a Gaussian reduced grid in ICMGGoifsINIT-file

| Variable short name | Variable WMO code | Description |
|---|---|---|
| stl1-4 | var139, 170, 183, 236 | Soil temperature level 1-4 |
| swvl1-4 | var39, 40, 41, 42 | Volumetric soil water layer 1-4 |
| sd | var141 | Snow depth |
| src | var198 | Skin reservoir content |
| skt | var235 | Skin temperature |
| ci | var31 | Sea ice area fraction |
| tsn | var238 | Temperature of snow layer |
| asn | var32 | Snow albedo |
| rsn | var33 | Snow density |
| sst | var34 | Sea surface temperature |
| istl1-4 | var35, 36, 37, 38 | Ice temperature layer 1-4 |
| chnk | var148 | Charnock |
| lsm | var172 | Land-sea mask |
| sr | var173 | Surface roughness |
| al | var174 | Albedo |
| sdor | var160 | Standard deviation of orography |
| isor | var161 | Anisotropy of sub-gridscale orography |
| anor | var162 | Angle of sub-gridscale orography |
| slor | var163 | Slope of sub-gridscale orography |
| lsrh | var234 | Logarithm of surface roughness length for heat |
| cvh | var28 | High vegetation cover |
| cvl | var27 | Low vegetation cover |
| tvh | var30 | Type of high vegetation |
| tvl | var29 | Type of low vegetation |
| sdfor | var74 | Standard deviation of filtered subgrid orography |
| alnid | var18 | Near IR albedo for diffuse radiation |





**Table A2.** Model level variables in spherical harmonics representation in ICMSHoifsINIT-file

| Variable short name | Variable WMO code | Description |
| --- | --- | --- |
| t | var130 | temperature |
| vo | var138 | vorticity |
| d | var155 | divergence |
| lnsp | var152 | logarithm of surface pressure |
| z | var129 | |

**Table A3.** Model level variables on a Gaussian reduced grid in ICMGGoifsINIUA-file

| Variable short name | Variable WMO code | Description |
| --- | --- | --- |
| q | var133 | specific humidity |
| clwc | var246 | specific cloud liquid water content |
| ciwc | var247 | specific cloud ice water content |
| cc | var248 | fraction of cloud cover |
| cswc | var76 | specific snow water content |
| crwc | var75 | specific rain water content |





**Appendix B:  Examples on manipulating files in spherical harmonics with CDO**

An example of workflow with CDO when constructing EDA only initial state perturbations to replace ICMSHoifsINIT. Note,
the field subtraction can be done in spherical harmonics representation as well.

1. Convert to grid point representation

    cdo -sp2gpl    ICMSHoifsINIT    gg_ctrl

    cdo -sp2gpl    pua_001    gg_eda+sv

    cdo -sp2gp    pert_001    gg_sv

2. Separate the variable fields (multi-field subtraction not supported)

    cdo -selvar,t    gg_eda+sv    gg_eda+sv_t

    cdo -selvar,d    gg_eda+sv    gg_eda+sv_d

    cdo -selvar,vo    gg_eda+sv    gg_eda+sv_vo

    cdo -selvar,lnsp    gg_eda+sv    gg_eda+sv_lnsp

cdo -selvar,z    gg_eda+sv    gg_eda+sv_z

    cdo -selvar,t    gg_sv    gg_sv_t

    cdo -selvar,d    gg_sv    gg_sv_d

    cdo -selvar,vo    gg_sv    gg_sv_vo

cdo -selvar,lnsp    gg_sv    gg_sv_lnsp

3. Change resolution of SV perturbations to match the other fields

    cdo -genbil,gg_eda+sv_t    gg_sv_t    grid

    cdo -remap,gg_eda+sv_t,grid    gg_sv_t    gg_sv_t_hr

    cdo -remap,gg_eda+sv_t,grid    gg_sv_d    gg_sv_d_hr

cdo -remap,gg_eda+sv_t,grid    gg_sv_vo    gg_sv_vo_hr

    cdo -remap,gg_eda+sv_t,grid    gg_sv_lnsp    gg_sv_lnsp_hr

4. Remove SVs

    cdo -sub gg_eda+sv_t    gg_sv_t_hr    gg_eda_t

    cdo -sub gg_eda+sv_d    gg_sv_d_hr    gg_eda_d

cdo -sub gg_eda+sv_vo    gg_sv_vo_hr    gg_eda_vo

    cdo -sub gg_eda+sv_lnsp    gg_sv_lnsp_hr    gg_eda_lnsp

5. Merge variables and transform back into spherical harmonics

    cdo -merge    gg_eda_t    gg_eda_vo    gg_eda_d    gg_eda_lnsp    gg_eda_z    gg_eda



```
          cdo -add    gg_ctrl    gg_eda    gg_final
385       cdo -gp2spl    gg_final    ICMSHoifsINIT
```

## Appendix C:  Applications in case studies: an example from forecasting a tropical cyclone

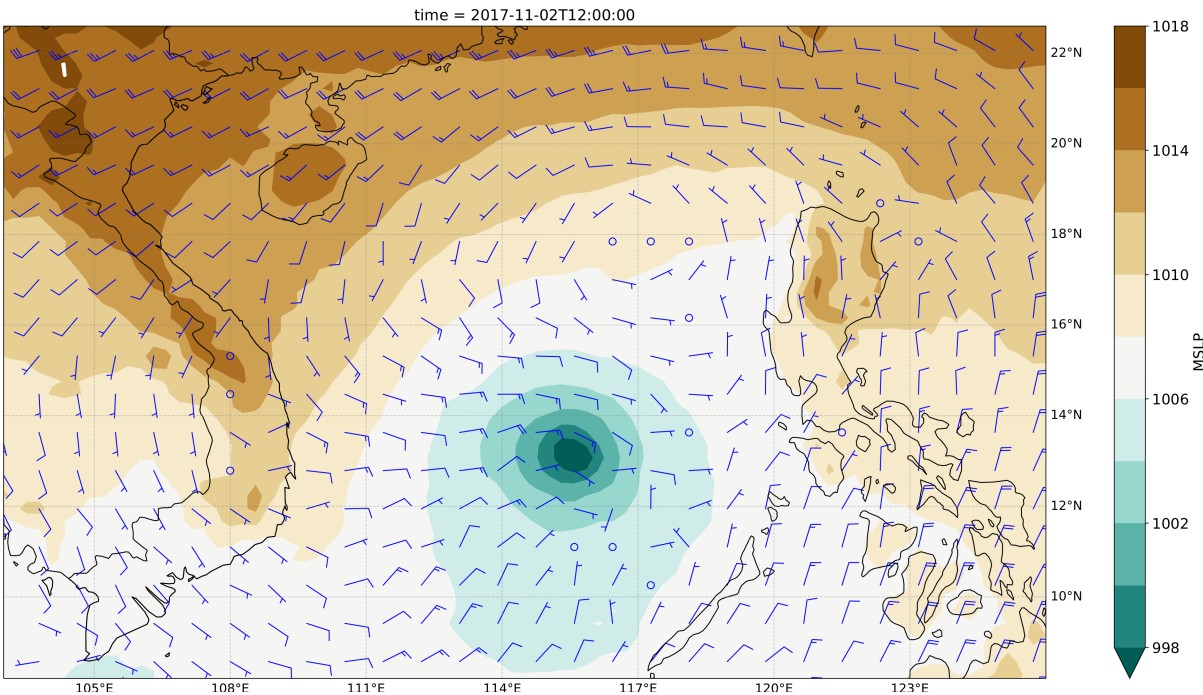

**Figure C1.** Ensemble mean initial state for MSLP (coloured contours) and 200 hPa wind vectors (blue barbs). $T_L 639$ resolution. November 2nd 12UTC.

*Author contributions.*  PO designed the workflow for reproducing the initial states, manages the repository of the initial states, coded the workflow manager OpenEPS and post-processing scripts used here. STKL was instrumental in helping designing the reproduction of the initial states. GDC provided crucial guidance in regards using OpenIFS for ensemble prediction purposes. LT and ME helped in designing and testing out the OpenEPS workflow manager. HJ was influential in creation of OpenEPS. All authors contributed to the writing of the manuscript.

*Competing interests.*  The authors declare that they have no conflict of interest.



*Acknowledgements.* The authors would like to thank ECMWF for helping in publishing this dataset. The authors are also grateful to CSC-IT Center for Science, Finland for providing computational resources, and Juha Lento at CSC-IT for helping in designing the workflow manager.

This work was supported by Academy of Finland (316939).



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
