# Peer review of "Ensemble prediction using a new dataset of ECMWF initial states -OpenEnsemble 1.0"

_Geoscientific Model Development, 2020_

## Referee Comment (RC1) · Anonymous Referee #1 · 27 Nov 2020

Review of

**gmd-2020-292: Ensemble prediction using a new dataset of ECMWF initial states - OpenEnsemble 1.0**

**by Pirkka Ollinaho, Glenn Carver , Simon Lang, Lauri Tuppi, Madeleine Ekblom, and Heikki Järvinen**

**1   Overview**

This paper introduces a data set of ECMWF ensemble initial conditions of three different resolutions for a period of one year (Dec 2016 to Nov 2017), which has been made freely available to the scientific community. Three different types of ensemble are made available – (1) a control member plus singular vector (SV) perturbations, (2) a control member plus ensemble of data assimilation (EDA) perturbations, and (3) a control member plus SV and EDA perturbations combined. Fifty ensemble members are available (plus the control members). The paper gives some explanation of how the data are accessed and manipulated using basic tools, and shows some examples of ensemble scores and a tropical cyclone example.

**2   Recommendation**

I would recommend that the paper is published in GMD as it highlights a very useful resource for atmospheric scientists. There are a number of changes that I would suggest to improve the clarity of the paper, as outlined below.

In my report the text of the paper is referenced by Lx (line x, as labeled in the manuscript), or by section/figure/equation/table number, and I often quote from the paper to help refer to the part that the comment refers to. Text that I suggest to be added are underlined, and items that I suggest to be removed are scored out.

**3   Scientific and major points**

1. L16-17 "Due to limitations in observations and in the data assimilation system, a measure of uncertainty remains in this state estimate.": Surely there is no direct measure of uncertainty from a single estimate.

2. L101-102: Could you please clarify which boundary conditions are perturbed in the EDA (e.g. sea surface temperatures, etc.)?

3. L108 "For each singular vector perturbation an individual linear combination from all singular vectors is constructed." I cannot make sense of this sentence. Should the first mention of "singular vector" in this sentence really be "ensemble of data assimilation perturbation"? That would then make sense – that each EDA perturbation is then complemented by a random combination of singular vectors.

4. L178 (step 3): Is this step specifically for the OpenIFS?

5. L237: "We use here operational ECMWF analyses from the forecast period as the truth." I understand that the states that are available in this OpenEnsemble 1.0 are also ECMWF *analyses*. Could the authors remind the reader of the difference between the mentioned ECMWF analyses and the states that are down-loadable as part of OpenEnsemble 1.0?

6. L238: "The model output is truncated to a $1° / 1°$ regular grid ..." Does this apply to the forecasts and the nominal 'truth' analysis? What are the three available model resolutions (and the operational analysis) in terms of degrees?

7. L258: "The recommended ensemble size was set to be four to eight members for scientific testing." All ensemble sizes are in this range or larger, so why aren't all the CRPS values in Fig. 3a equal to their 'fair' values?

8. L269-270: "Noticeably, $T_L159$ resolution with EDA perturbations scores better in the Tropics than $T_L639$ with only SV perturbations active." This seems to be true only for $t < \sim 168$h.

9. L270-271: "The SV perturbations in the Tropics consist only of perturbations around active Tropical Cyclones, thus the relatively high fair-CRPS in the Tropics is expected."

   (a) This means that the authors have used the five SVs that are produced associated with tropical cyclones (as mentioned on L92), instead of the 50 NH/SH SVs produced. How can one use these five tropical cyclone SVs from the downloaded data (instead of the 50 NH/SH ones)?

   (b) "... relatively high fair-CRPS ..." Relatively high compared to what (especially as the fair CRPS values for the tropics are smaller than those for the NH)?

   (c) L276-277: "In TR the increase of skill becomes noticeable beyond forecast lead time of 96 h (120 h) for EDA (SV) perturbations." This is quite a subjective measure of the time that the factor-1 and factor-1.2 lines split. For example you could argue that for the SV perturbations the factor-1 and factor-1.2 lines split between 24 and 48 hours into the forecasts. It really depends on how much one zooms into the plots.

10. Are there any plans to include other years' data.

**4 Presentational points**

1. L102: Please define SPPT.

2. L135: Are these URLs stored in a permanent repository? Is there a parent web page that has a click-able list all of the files?

3. L136-137: What do pan, psu, pua and pert mean? See also point 5 below.

4. L143-144: "On top of these four files, OpenIFS requires static climatological files for radiation calculations and various namelists describing the hybrid sigma coordinates etc." Are these files and namelists also available from the repository? If not, does this mean that OpenIFS cannot be run?

5. Table 1 is a little mysterious to me. I don't know what the "Use as" column means – it appears to be just a list of files and empty elements. Also some of the files do not have a description. The caption of this table would benefit from some more explanation, e.g. enabling the reader to understand the filenames. See also point 3 above.

6. Similarly for Table 2, the "Manipulation" column is a little mysterious to me. I would have thought that this column would contain references to some procedures that need to be performed to manipulate the files. Instead it just contains a list of files in addition to those listed in the "Files needed" column.

7. All figures: Would it be possible to include the key to the lines in both panels? When I read the paper I had to zoom-into the plots a lot and it would be useful to have the keys right next to the lines themselves in all the panels.

8. L256: Would the authors consider defining the CRPS mathematically?

9. L264: This sounds confusing. Should it read, "As per construction, the  smaller the normal CRPS score, the  more ensemble members the ensemble contains (Fig 3)."?

10. L295-296: "Lang et al. (2012) show how especially the SV perturbations can rapidly alter the TC location and intensity." This reads that Lang et al. (2012)'s study is of the same tropical cyclone. This cannot obviously be the case as typhoon Damrey happened after this publication. Would suggest that the sentence reads, ""Lang et al. (2012) show how especially the SV perturbations can rapidly alter  a TC's location and intensity."

11. L319-320: Suggested change, "Increasing the amplitudes of the initial state perturbations result in an increase of the forecast skill of the system, demonstrating that inflation tuning of the initial conditions can improve forecast skill."

12. Appendix B:

    (a) Are these commands performed in the shell or inside another piece of software?
    (b) Most of the commands have an obvious syntax, but some generic descriptions would help (e.g. for one of the more obvious commands: cdo -sp2gpl  <output file>).
    (c) I would recommend repeating the definition of "CDO" and the reference to CDO documentation in this appendix.
    (d) Where can CDO be obtained from?
    (e) What is the difference between -genbil and -remap?

13. Appendix C: This figure would be more logically placed in the main text instead of in a separate appendix. Perhaps there is a reason for putting it separately?

**5 Minor points (typos, etc)**

1. L129: "... process and  the number of observations ..."

2. L157: "... the +/- symmetry is  built into the files ..."

3. L181: "... tasks  e.g. ..."

4. L213: Also, the different initial perturbation types (SV and EDA) were tested separately and together in order to illustrate ...

5. L216: "Running large numbers of ensemble forecasts requires  substantial  computational resources."

6. L243: "Both kinds of perturbation also improve ..."

7. L256: "Leutbecher et al. (2017) illustrated ..."

8. L262: "from 2019 onwards in the ECMWF operational ensemble ..."

9. L268: "... this is  in line with ..."

10. L286: "... west of the Philippines ..."

11. L304: "... momentum exchange between the ocean surface and the atmosphere?"

12. L306: " ... states covering a one year period ..."

13. L324: "In operational ensemble configurations, having ..."

14. L328: "... scheme (Buizza et al., 2008a)  and also an early version ..."

15. Table A2: I presume that $z$ is the geopotential, which is missing from the description.

16. L 355: "... field subtraction can be done in spherical harmonics representation as well as in grid-point space."

---

## Referee Comment (RC2) · Hannah Christensen (Referee) · 18 Dec 2020

Ensemble prediction using a new dataset of ECMWF initial states - OpenEnsemble 1.0

Pirkka Ollinaho , Glenn D. Carver , Simon T. K. Lang , Lauri Tuppi , Madeleine Ekblom, and Heikki Järvinen

This paper describes a new dataset of perturbed initial conditions which enables researchers to initialise ensemble forecasts using the OpenIFS. The initial condition perturbation methodology is that used operationally (a couple of cycles ago) at ECMWF. The authors also present and provide a workflow manager to assist in producing ensemble forecasts using these initial condition files. Results are presented which demonstrate the impact of different initial condition perturbations available in the

dataset on forecast skill.

I was happy to review this manuscript, as I believe the dataset and workflow manager described within will be a real boon for the research community. I recommend the manuscript be published subject to the correction suggested below, which I hope will improve the clarity of the manuscript.

» General comments

1. Several of your figures show subtle differences between the forecast skill with EDA, SV or both perturbations. It would be helpful to assess the statistical significance of the differences between different initialisation methods. This will demonstrate that the database contains sufficient independent start dates for researchers' needs. For the significance you will need to compare forecasts pairwise for the same dates, as some dates will be more predictable than others. You could include the 95% significance levels on your scores in the figures (e.g. as in Christensen et al, 2017, DOI:10.1002/qj.3075), though I appreciate this may be difficult for the figures showing results from many experiments.

2. Do you plan to make more start dates available, and with what kind of frequency?

» Specific comments

L21-24 While it is harder for the academic community to contribute to ensemble forecasting research at the moment, it is not impossible. But it does have to be carried out in close collaboration with an operational centre, e.g. through an ECMWF special project. You should soften this statement to acknowledge this.

L45-48 "Although the . . . quality early on". This statement doesn't seem to fit here. You could move it up to the end of paragraph L32-36

L73 "physical grid" – you should expand this for those unfamiliar with the IFS, to indicate you mean the grid on which the model physical parametrisations are run

L92 & L106 On the first line, 50 SVs are mentioned, while on the second, 25 SVs are generated. This needs to be clarified.

L124 clarify, e.g. to: "the highest model resolution available in OpenEnsemble is TL639 (instead of TCO639 as used at ECMWF)"

L125 This seems to be a good place to highlight the three resolutions available in OpenEnsemble, which I don't think you mention explicitly in this section at the moment

L137-152 including Table 1. This was an extremely confusing section. - what are "pan", "psu", "pua" and "pert" on L137 - L139-140 highlight briefly why you need files on both GG and SH grids (different variables are available on different grids, as expanded in the appendix) - the file names are long and unintuitive – what do all the bits of the filename mean? ICM? UA vs T? There doesn't seem to be an indicator of whether the files are model levels or surface levels in the name. - Table 1 is very perplexing. What does "use as" mean? - descriptions are missing for rows 5, 6, 7 in table 1 - the contents of the CL file are not included in a table in the Appendix. This would be helpful - while running the model, is the assumption that all the surface fields are held constant, or do they evolve? I suppose the land model is included, so those variables would evolve, but what about SST? - L147-152 I was confused why some of the files contained only EDA perturbations, while others contained EDA plus SV. Again the filenames "pan", "psu", "pua" and "pert" are rather unintuitive so don't help the reader understand what's going on - what has become apparent by this point is that perhaps the EDA and SV perturbations are not applied to all the variables in the different input files needed by the IFS. It would be helpful to explicitly say which variables are perturbed by each method in section 3.3

L154-160 and Table 2. I found it helpful for you to include the cdo commands, and this does make sure that the user is absolutely clear as to how to use the data. However I found Table 2 confusing. In particular the third column "manipulation" was a mystery.

L244-245 Not only do they not grow, they shrink over the first 48 hours. Why is this?

[Figure]

Can you comment here? In addition, the initial spread seems to be too large when using the EDA perturbations – again, why is this?

L264-265 "the normal CRPS . . .(Fig 3)" – this sentence needs to be rephrased – perhaps too many "the"s

L268 "generates less spread than using only EDA perturbations" this figure is not showing spread and error, but is showing CRPS, so this statement should be linked through to a resultant lower CRPS.

L327 Is the backscatter scheme not also available, e.g. as legacy code?

Warmest wishes, Hannah Christensen
* * *

---

## Author Comment (AC1) · 8 Feb 2021

Response to Referee #1

We want to thank Referee #1 for the insightful and helpful comments to improve the manuscript. We feel the manuscript was much improved after reviewing it based on these comments. We have answered to the comments below point by point.

We want to mention here that even though it was not pointed out by either of the Referees, we have decided to include the wave model (WAM) initial states to this release version of the OpenEnsemble dataset. We think this addition will be beneficial for the research community since OpenIFS is recommended to be run with WAM enabled. We are currently running an additional research experiment with TL159 resolution in which WAM is enabled on top of the EDA and SV perturbation. The results are to be included in Fig 5 along with some text additions and modifications.

*3 Scientific and major points*

*1. L16-17 "Due to limitations in observations and in the data assimilation system, a mea-sure of uncertainty remains in this state estimate.": Surely there is no direct measure of uncertainty from a single estimate.*

Yes, a single estimate cannot be used to estimate uncertainty. "a measure" was used here without implying anything about the quantity of the uncertainty, but we recognize it leaves some room for interpretation.

**We have changed the text to:** "…system, an unknown amount of uncertainty remains..."

*2. L101-102: Could you please clarify which boundary conditions are perturbed in the EDA (e.g. sea surface temperatures, etc.)?*

Only SSTs are perturbed. **Text changed to:** "...observations and sea surface temperatures are perturbed..."

*3. L108 "For each singular vector perturbation an individual linear combination from all sin-gular vectors is constructed." I cannot make sense of this sentence. Should the first mention of "singular vector" in this sentence really be "ensemble of data assimilation perturbation"? That would then make sense - that each EDA perturbation is then complemented by a random combination of singular vectors.*

The sentence is referring to how each of the 25 unique singular vector perturbations are constructed.

**We have clarified the sentence:** "Each of the 25 singular vector perturbations is constructed through a linear combination of the leading singular vectors which are calculated separately for NH/TR/SH. As such, each of the 25 singular vector perturbations contain some form of the calculated leading singular vectors in all NH/TR/SH. This is done by scaling the leading singular vectors by random numbers drawn from a multi-variate Gaussian distribution (see Leutbecher and Palmer, 2008; Leutbecher and Lang, 2014)."

*4. L178 (step 3): Is this step specifically for the OpenIFS?*

Yes, it is since no other model infrastructure is included in OpenEPS.

**We have changed the text accordingly:** "3. Run OpenIFS model forecasts..."

*5. L237: "We use here operational ECMWF analyses from the forecast period as the truth."*
*I understand that the states that are available in this OpenEnsemble 1.0 are also ECMWF*
*analyses . Could the authors remind the reader of the difference between the mentioned*
*ECMWF analyses and the states that are down-loadable as part of OpenEnsemble 1.0?*

Thank you for pointing this out. The two are indeed basically the same, but the ECMWF analyses are available on 6 hourly intervals. There is a model (IFS) version difference between the operational analyses and OpenEnsemble: ECMWF analyses from 1$^{st}$ Dec 2016 to 10$^{th}$ July 2017 use IFS CY43r1, the analyses covering 11$^{th}$ July to 30$^{th}$ November 2017 use IFS CY43r3. However, since we are generating the forecasts with an even older model version, we do not think this matters in regards to what is discussed in our manuscript. But, we do agree this should be discussed in the manuscript.

**The text now includes a description of the differences:** "We use here operational ECMWF analyses truncated to a 1/1 regular grid from the forecast period as the truth. The operational analyses are available on 6 hourly interval instead of the 12 hourly interval available through OpenEnsemble 1.0. We want to note that the operational ECMWF analyses covering the same time period as OpenEnsemble 1.0 use two different IFS versions: CY43r1 is used until 10$^{th}$ July 2017 and CY43r3 from 11$^{th}$ July onwards. Thus even when using OpenIFS version CY43r3 to generate forecasts from OpenEnsemble 1.0, verification scores calculated with operational ECMWF analyses might appear somewhat degraded due to differences in the underlying model version."

*6. L238: "The model output is truncated to a 1/1 regular grid ..." Does this apply to the*
*forecasts and the nominal `truth' analysis? What are the three available model resolutions*
*(and the operational analysis) in terms of degrees?*

Yes, the truncation applies to both forecast and analysis states (see text changes to the previous comment). The model resolutions in degrees at equator are: ~0.28/~0.28 (TL639), 0.45/0.45 (TL399), 1.125/1.125 (TL159). We recognize a mistake in the used vocabulary, obviously the word "truncation" applies only to the two higher resolutions.

**We have changed the text:** "The model output is interpolated to a 1/1 regular grid..."

*17. L258: "The recommended ensemble size was set to be four to eight members for scientific*
*testing." All ensemble sizes are in this range or larger, so why aren't all the CRPS values*
*in Fig. 3a equal to their `fair' values?*

Fig. 3a shows only normal CRPS values for different ensemble sizes, whereas only fair values are shown in Fig. 3b. We understand the text needs clarification and also noticed that the Fig. 3b y-title is "CRPS" instead of "FAIR CRPS".

**We have clarified the text and changed the y-title:** "Fig 3a shows normal CRPS calculated for ensembles of various sizes. Fig 3b shows the fair-CRPS values for the same experiments."

*8. L269-270: "Noticeably, TL159 resolution with EDA perturbations scores better in the Tropics than TL639 with only SV perturbations active." This seems to be true only for t <~ 168h .*

We have done some more in-detail comparison of some of the experiments on the request of the other Referee. There are now two additional panels in Figures 4 and 5 showing pairwise fair-CRPS differences between a reference experiment and selected experiments.

**Changes to the text :** "Noticeably, TL159 resolution with EDA perturbations scores better in the Tropics than TL639 with only SV perturbations active for forecast lead times shorter than about 168 h."

*9. L270-271: "The SV perturbations in the Tropics consist only of perturbations around active Tropical Cyclones, thus the relatively high fair-CRPS in the Tropics is expected."*

*(a) This means that the authors have used the five SVs that are produced associated with tropical cyclones (as mentioned on L92), instead of the 50 NH/SH SVs produced. How can one use these five tropical cyclone SVs from the downloaded data (instead of the 50 NH/SH ones)?*

We hope this was clarified in comment 3.3. (the TC SVs are already included in each SV perturbation). **No further action taken.**

*(b) "... relatively high fair-CRPS ..." Relatively high compared to what (especially as the fair CRPS values for the tropics are smaller than those for the NH)?*

This should read "higher than EDA".

**Text has been changed accordingly:** "… . Therefore, the SV perturbations resulting in a higher fair-CRPS than the EDA perturbations in the Tropics is expected."

*(c) L276-277: "In TR the increase of skill becomes noticeable beyond forecast lead time of 96 h (120 h) for EDA (SV) perturbations." This is quite a subjective measure of the time that the factor-1 and factor-1.2 lines split. For example you could argue that for the SV perturbations the factor-1 and factor-1.2 lines split between 24 and 48 hours into the forecasts. It really depends on how much one zooms into the plots.*

Yes, you are correct. The similarities/differences are better shown in the new figures. Copy-paste from our answer to Referee #2:

**The chapter related to Fig 4 now reads:**
Figure 4 (a,b) shows fair-CRPS for all the experiments using standard initial state perturbation amplitudes. Additionally, Fig 4 (c,d) shows pairwise fair-CRPS differences with 95% confidence intervals (calculated via bootstrapping) between TL639EDA+SV experiment and selected experiments. Using only SV perturbations less skillful ensembles than when using only EDA perturbations, this is inline with findings in Buizza et al. (2008a) and Lang et al. (2012). Noticeably, TL159 resolution with EDA perturbations scores better in the Tropics than TL639 with only SV perturbations active for forecast lead times shorter than about 168 h. The SV perturbations in the Tropics consist only of perturbations around active Tropical Cyclones, therefore the SV perturbations resulting in a higher fair-CRPS than the EDA perturbations in the Tropics is expected.

Nonetheless, having SV perturbations active on top of EDA perturbations improves the forecast skill in both NH and TR. Interestingly, when comparing the EDA+SV experiments for the different resolutions, TL159 scores the best while TL639 scores the worst for the 0th time step of the model integration. This is likely caused by analysis and forecast model version differences being smoothed out more in the lower resolution experiments than in the higher resolution experiments.

**The chapter related to Fig 5 now reads:**
The SV perturbation amplitude is a tuning parameter of the ensemble (Leutbecher and Palmer, 2008; Leutbecher and Lang,2014). Figure 5 illustrates the sensitivity of TL159 resolution ensemble to a change in the initial perturbation amplitude. There is a noticeable increase of skill beyond a 12 h forecast lead time in NH when increasing the EDA perturbation amplitudes by a factor of 1.2. Increasing the SV perturbation amplitudes result in an increase in skill for all forecast lead times. In TR the increase of skill becomes noticeable beyond forecast lead time of 12 h (48 h) for EDA (SV) perturbations.

*10. Are there any plans to include other years' data.*

Not currently, this depends entirely on how popular the dataset will become and what would be the community demand for a dataset covering more years and/or resolutions. **No action taken.**

*4 Presentational points*
*1. L102: Please define SPPT.*

**Done.**

*2. L135: Are these URLs stored in a permanent repository? Is there a parent web page that has a click-able list all of the files?*

There is no such webpage. We trust the potentially more active users will code their own bash (etc.) scripts to download a (sub)set of the files. **No action taken.**

*3. L136-137: What do pan, psu, pua and pert mean? See also point 5 below.*

**Please see answer to 6.**

*4. L143-144: "On top of these four files, OpenIFS requires static climatological files for radiation calculations and various namelists describing the hybrid sigma coordinates etc."*
*Are these files and namelists also available from the repository? If not, does this mean that OpenIFS cannot be run?*

All the necessary files to run openifs are now either provided via OpenEnsemble or openly available from an ftp-server: ftp://ftp.ecmwf.int/pub/openifs/ifsdata

**Please see answer to 6.**

*5. Table 1 is a little mysterious to me. I don't know what the "Use as" column means - it appears to be just a list of files and empty elements. Also some of the files do not have*

*a description. The caption of this table would benefit from some more explanation, e.g.
enabling the reader to understand the filenames. See also point 3 above.*

**Please see answer to 6.**

*6. Similarly for Table 2, the "Manipulation" column is a little mysterious to me.
I would have thought that this column would contain references to some procedures that need to
be performed to manipulate the files. Instead it just contains a list of files in addition to
those listed in the "Files needed" column.*

The other Referee also commented about hard readability of Section 4. We agree that the Section
was difficult to understand and that the tables required some deciphering skills to be understood.

**We have now reorganized and reworded most of the Section, including the tables. We have
also added a new Table describing what initial files are required to run OpenIFS.**

*7. All figures: Would it be possible to include the key to the lines in both panels? When I
read the paper I had to zoom-into the plots a lot and it would be useful to have the keys
right next to the lines themselves in all the panels.*

**The legend box is now included in both panels of the Figures.**

*8. L256: Would the authors consider defining the CRPS mathematically?*

We trust the reader will find more than enough information from the cited papers, in particular
Leutbecher, 2017. **No action taken.**

*9. L264: This sounds confusing. Should it read, "As per construction, the
smaller the normal CRPS score, the better the more ensemble members the ensemble contains (Fig
3)."?*

**Changed to:** "As per construction, the regular CRPS is lower (i.e. better) for an ensemble with
more members than for an ensemble with fewer members (Fig 3)." **Also, we have changed
"normal CRPS" to "regular CRPS" throughout the text.**

*10. L295-296: "Lang et al. (2012) show how especially the SV perturbations can rapidly alter
the TC location and intensity." This reads that Lang et al. (2012)'s study is of the same
tropical cyclone. This cannot obviously be the case as typhoon Damrey happened after
this publication. Would suggest that the sentence reads, ""Lang et al. (2012) show how
especially the SV perturbations can rapidly alter  a TC's location and intensity."*

Yes, we are indeed referring to how the SVs function in general. **The text is changed as suggested.**

*11. L319-320: Suggested change, "Increasing the amplitudes of the initial state perturbations
result in an increase of the forecast skill of the system, demonstrating that inflation tuning
of the initial conditions can improve forecast skill."*

Thank you for the suggestion, **we have included it with a minor modification:** "...can improve probabilistic skill."

*12. Appendix B:*
*(a) Are these commands performed in the shell or inside another piece of software?*

In the shell, **we have included this point in:** "An example of workflow in Bash shell with Climate Data Operators (CDO; Schulzweida, 2019; available at https://code.mpimet.mpg.de/projects/cdo)..."

*(b) Most of the commands have an obvious syntax, but some generic descriptions would help (e.g. for one of the more obvious commands: cdo -sp2gpl  <output file>).*

Thank you for the suggestion, **a generic function description is now included**:

1. Convert to grid point representation (cdo -sp2gpl  <output file>)

cdo -sp2gpl ICMSHoifsINIT gg_ctrl
…
2. Separate the variable fields (multi-field subtraction not supported) (cdo -selvar,  <output file>)
...
3. Change resolution of SV perturbations to match the other fields (cdo -genbil,<grid>  <output file> is used the first generate interpolation weights; cdo -remap,<grid>,<weights>  <output file> then does the interpolation applying these weights.)
…

*(c) I would recommend repeating the definition of "CDO" and the reference to CDO documentation in this appendix.*

Thank you for suggestion, **this is now included. Please, see answer to (a).**

*(d) Where can CDO be obtained from?*

**Please, see answer to (a).**

*(e) What is the difference between -genbil and -remap?*

-genbil is used to generate bilinear interpolation weights, -remap applies these weights to the fields and does the interpolation. **Please, see answer to (b).**

*13. Appendix C: This figure would be more logically placed in the main text instead of in a separate appendix. Perhaps there is a reason for putting it separately?*

We feel the main manuscript already is quite heavy on the figures. The additional information provided by Fig C1 is, in our view, only valuable to a specific group of readers. Thus it was placed in the Appendix section. **No action taken.**

*5 Minor points (typos, etc)*

Thank you for the suggestions, we agree to these changes (with the exception of point 7). **Text corrected.**

*1. L129: "... process and  the number of observations ..."*

*2. L157: "... the +/- symmetry is  built into the files ..."*

*3. L181: "... tasks  e.g. ..."*

*4. L213: Also, the different initial perturbation types (SV and EDA) were tested separately and together in order to illustrate ...*

*5. L216: "Running large numbers of ensemble forecasts requires  substantial  computational resources."*

*6. L243: "Both kinds of perturbation also improve ..."*

*7. L256: "Leutbecher et al. (2017) illustrated ..."*

This is a single writer paper. **No action taken.**

*8. L262: "from 2019 onwards in the ECMWF operational ensemble ..."*

*9. L268: "... this is  in line with ..."*

*10. L286: "... west of the Philippines ..."*

*11. L304: "... momentum exchange between the ocean surface and the atmosphere?"*

*12. L306: " ... states covering a one year period ..."*

*13. L324: "In operational ensemble configurations, having ..."*

*14. L328: "... scheme (Buizza et al., 2008a)  and also an early version ..."*

*15. Table A2: I presume that z is the geopotential, which is missing from the description.*

*16. L 355: "... field subtraction can be done in spherical harmonics representation as well as in grid-point space."*

**Response to Hannah Christensen (Referee #2)**

We want express our gratitude to Hannah Christensen for the generous words describing the dataset and for the insightful review of the manuscript. We feel the content of the manuscript and the flow of the text have been improved as a result. Our point by point response to the comments can be found below.

We want to mention here that even though it was not pointed out by either of the Referees, we have decided to include the wave model (WAM) initial states to this release version of the OpenEnsemble dataset. We think this addition will be beneficial for the research community since OpenIFS is recommended to be run with WAM enabled. We are currently running an additional research experiment with TL159 resolution in which WAM is enabled on top of the EDA and SV perturbation. The results are to be included in Fig 5 along with some text additions and modifications.

*» General comments*
*1. Several of your figures show subtle differences between the forecast skill with EDA, SV or both perturbations. It would be helpful to assess the statistical significance of the differences between different initialisation methods. This will demonstrate that the database contains sufficient independent start dates for researchers' needs. For the significance you will need to compare forecasts pairwise for the same dates, as some dates will be more predictable than others. You could include the 95% significance levels on your scores in the figures (e.g. as in Christensen et al, 2017, DOI:10.1002/qj.3075), though I appreciate this may be difficult for the figures showing results from many experiments.*

Thank you for the suggestion, **we have added pairwise fair-CRPS comparisons as additional panels to Figures 4 and 5.**

**The chapter related to Fig 4 now reads:**
Figure 4 (a,b) shows fair-CRPS for all the experiments using standard initial state perturbation amplitudes. Additionally, Fig 4 (c,d) shows pairwise fair-CRPS differences with 95% confidence intervals (calculated via bootstrapping) between TL639EDA+SV experiment and selected experiments. Using only SV perturbations less skillful ensembles than when using only EDA perturbations, this is inline with findings in Buizza et al. (2008a) and Lang et al. (2012). Noticeably, TL159 resolution with EDA perturbations scores better in the Tropics than TL639 with only SV perturbations active for forecast lead times shorter than about 168 h. The SV perturbations in the Tropics consist only of perturbations around active Tropical Cyclones, therefore the SV perturbations resulting in a higher fair-CRPS than the EDA perturbations in the Tropics is expected. Nonetheless, having SV perturbations active on top of EDA perturbations improves the forecast skill in both NH and TR. Interestingly, when comparing the EDA+SV experiments for the different resolutions, TL159 scores the best while TL639 scores the worst for the 0th time step of the model integration. This is likely caused by analysis and forecast model version differences being smoothed out more in the lower resolution experiments than in the higher resolution experiments.

**The chapter related to Fig 5 now reads:**
The SV perturbation amplitude is a tuning parameter of the ensemble (Leutbecher and Palmer, 2008; Leutbecher and Lang,2014). Figure 5 illustrates the sensitivity of TL159 resolution ensemble to a change in the initial perturbation amplitude. There is a noticeable increase of skill beyond a 12 h forecast lead time in NH when increasing the EDA perturbation amplitudes by a factor of 1.2. Increasing the SV perturbation amplitudes result in an increase in skill for all forecast lead times. In

TR the increase of skill becomes noticeable beyond forecast lead time of 12 h (48 h) for EDA (SV) perturbations.

*2. Do you plan to make more start dates available, and with what kind of frequency?*

Not currently. This depends entirely on how popular the dataset will become and what would be the community demand for a dataset covering more years and/or resolutions. **No action taken.**

*» Specific comments*
*L21-24 While it is harder for the academic community to contribute to ensemble fore-casting research at the moment, it is not impossible. But it does have to be carried out in close collaboration with an operational centre, e.g. through an ECMWF special project. You should soften this statement to acknowledge this.*

Yes, you are correct. **We have modified the abstract as well and changed the text to:** "...ensemble forecasting research has been mostly a duty of major operational forecasting centres or academic institutions closely collaborating with said centres."

*L45-48 "Although the . . . quality early on". This statement doesn't seem to fit here. You could move it up to the end of paragraph L32-36*

**Done.**

*L73 "physical grid" – you should expand this for those unfamiliar with the IFS, to indicate you mean the grid on which the model physical parametrisations are run*

**We have changed the text to:** "...describe the layout of the grid-points used to compute, for example, the physical parameterization terms:"

*L92 & L106 On the first line, 50 SVs are mentioned, while on the second, 25 SVs are generated. This needs to be clarified.*

Referee #1 also wrote this was not clear, copy-paste from our response to Referee #1:

**We have clarified the sentence:** "Each of the 25 singular vector perturbations is constructed through a linear combination of the leading singular vectors which are calculated separately for NH/TR/SH. As such, each of the 25 singular vector perturbations contain some form of the calculated leading singular vectors in all NH/TR/SH. This is done by scaling the leading singular vectors by random numbers drawn from a multi-variate Gaussian distribution (see Leutbecher and Palmer, 2008; Leutbecher and Lang, 2014)."

*L124 clarify, e.g. to: "the highest model resolution available in OpenEnsemble is TL639 (instead of TCO639 as used at ECMWF)"*

Thank you for the suggestion, **please see the answer below.**

*L125 This seems to be a good place to highlight the three resolutions available in OpenEnsemble, which I don't think you mention explicitly in this section at the moment*

Thank you for the suggestion, **the text is changed to:** "We use IFS cycle 43R3, which was operational from July 11th 2017 to June 5th 2018, to generate the ensemble initial states. This model version is also the basis for the OpenIFS release CY43R3v1. The single major difference to the operational ensemble setup is that the highest model resolution available in OpenEnsemble is TL639 (~32km) (instead of TCO639; ~18km). The other available resolutions provided in OpenEnsemble are TL399 (～50km) and TL159 (～120km).

*L137-152 including Table 1. This was an extremely confusing section. - what are "pan", "psu", "pua" and "pert" on L137 - L139-140 highlight briefly why you need files on both GG and SH grids (different variables are available on different grids, as expanded in the appendix) - the file names are long and unintuitive – what do all the bits of the filename mean? ICM? UA vs T? There doesn't seem to be an indicator of whether the files are model levels or surface levels in the name. - Table 1 is very perplexing. What does "use as" mean? - descriptions are missing for rows 5, 6, 7 in table 1 - the contents of the CL file are not included in a table in the Appendix. This would be helpful - while running the model, is the assumption that all the surface fields are held constant, or do they evolve? I suppose the land model is included, so those variables would evolve, but what about SST? - L147-152 I was confused why some of the files contained only EDA perturbations, while others contained EDA plus SV. Again the filenames "pan", "psu", "pua" and "pert" are rather unintuitive so don't help the reader understand what's going on - what has become apparent by this point is that perhaps the EDA and SV perturbations are not applied to all the variables in the different input files needed by the IFS. It would be helpful to explicitly say which variables are perturbed by each method in section 3.3*

*L154-160 and Table 2. I found it helpful for you to include the cdo commands, and this does make sure that the user is absolutely clear as to how to use the data. However I found Table 2 confusing. In particular the third column "manipulation" was a mystery.*

The other Referee also commented about hard readability of Section 4. We agree that the Section was difficult to understand and that the tables required some deciphering skills to be understood.

**We have now reorganized and reworded most of the Section, including the tables. We have also added a new Table describing what initial files are required to run OpenIFS.**

*L244-245 Not only do they not grow, they shrink over the first 48 hours. Why is this? Can you comment here? In addition, the initial spread seems to be too large when using the EDA perturbations – again, why is this?*

We believe the decay seen in the experiments where EDA perturbations are included is caused because we:

1) re-centre 6-hour forecast EDA perturbations on the initial analysis instead of starting directly from the EDA member. This artificially increases the small scale variability of the perturbed initial states which subsequently decays.

* see Lang, S.T.K., Bonavita, M. and Leutbecher, M. (2015), On the impact of re-centring initial conditions for ensemble forecasts. Q.J.R. Meteorol. Soc., 141: 2571-2581. https://doi.org/10.1002/qj.2543

2) do not run with a model uncertainty representation which has a large impact on spread growth in the tropics.

As for the too large initial spread, we believe this is mainly caused by differences in the analysis and forecast model versions: TL639 ENS has more small scale structures which are somewhat different from the analysis against we verify. In TL159 these are smoothed out due to the coarser resolution and hence the model scores better. **We have included the following sentence to discuss this (see answer to 1 for the context):** "This is likely caused by analysis and forecast model version differences being smoothed out more in the lower resolution experiments than in the higher resolution experiments."

*L264-265 "the normal CRPS . . .(Fig 3)" – this sentence needs to be rephrased – perhaps too many "the"s*

**Changed to:** "As per construction, the regular CRPS is lower (i.e. better) for an ensemble with more members than for an ensemble with fewer members (Fig 3)." **Also, we have changed "normal CRPS" to "regular CRPS" throughout the text.**

*L268 "generates less spread than using only EDA perturbations" this figure is not showing spread and error, but is showing CRPS, so this statement should be linked through to a resultant lower CRPS.*

**Corrected:** "Using only SV perturbations results in less skillful ensembles than when using only EDA perturbations,..."

*L327 Is the backscatter scheme not also available, e.g. as legacy code?*

Thank you for pointing this out, SKEB is indeed included as a legacy code in OpenIFS. **We have now mentioned this in the discussion section:** "The OpenIFS release based on IFS CY43R3 includes the stochastically perturbed parameterization tendencies (SPPT) scheme (Buizza et al., 2008a), the *Stochastic Kinetic Energy Backscatter (SKEB; Berner et al., 2009)* and also an early version..."